# An Internet of Things-Based Production Scheduling for Distributed Two-Stage Assembly Manufacturing with Mold Sharing

Yin Liu [1,2,†], Cunxian Ma [1,†] and Yun Huang [1,*]

1 School of Business, Macau University of Science and Technology, Macao 999078, China
2 School of Electronic Engineering, Huainan Normal University, Huainan 232038, China
* Correspondence: yuhuang@must.edu.mo
† The authors contribute equally to the paper.

**Abstract:** In digital product and ion scheduling centers, order–factory allocation, factory–mold allocation, and mold routing can be performed centrally and efficiently to maximize the utilization of manufacturing resources (molds). Therefore, in this paper, a manufacturing resource (molds)-sharing mechanism based on the Internet of Things (IoT) and a cyber-physical production system (CPPS) is designed to realize the coordinated allocation of molds and production scheduling. A mixed-integer mathematical model is developed to optimize the cost structure and obtain a reasonable profit solution. A heuristic algorithm based on evolutionary reversal is used to solve the problem. The numerical results show that based on the digital coordinated production scheduling method, distributed two-stage assembly manufacturing with shared molds can effectively reduce the order delay time and increase potential benefits for distributed production enterprises.

**Keywords:** order allocation; mold sharing; distributed manufacturing; IoT; CPPS





## 1. Introduction

The majority of businesses in the discrete manufacturing sector use distributed two-stage assembly flow-shop scheduling. Components are initially processed in the processing step before being assembled into products in the final assembly stage [1]. How to coordinate the two-stage production plan to ensure the timely delivery of orders and improve the comprehensive utilization of manufacturing resources (molds) is the main problem in the field of distributed two-stage assembly scheduling. In recent years, the research on the distributed two-stage assembly scheduling problem (DTASP) has been increasing gradually in the academic field, and some research results have been achieved. However, there is a small amount of the literature on how to improve the comprehensive utilization rate of manufacturing resources (molds) in distributed manufacturing environments.

For the sharing and scheduling of manufacturing resources (molds) in a distributed production environment, Raa et al. [2] study three main problems in the supply chain, namely production scheduling, mold transferring and transportation, and distribution. Two mathematical models for minimizing the moving cost of molds are proposed (i.e., model 1 assumes that the time and cost of moving molds between different factories are different, and model 2 assumes that the time and cost of moving molds between different factories are the same), as well as a mixed-integer linear programming model considering the production cost, transportation cost, warehouse purchase/shipment processing cost, and storage cost. Finally, it is confirmed that mold sharing can reduce the total production cost by about 10%. In the existing literature on the DTASP, most of them take the minimization of the completion time, the minimization of the average completion time, or the weighted sum of both as the objectives and consider the cost minimization

problem. For example, Yavari et al. [3] consider the total weighted delay cost minimization when studying the order acceptance and scheduling (OAS) problem with the total weighted delay as the scheduling criterion. However, there is little research on the relationship between mold sharing and enterprises' overall profit maximization in distributed manufacturing environments.

Therefore, this paper takes the distributed manufacturing operation of a large household appliance manufacturing enterprise (a typical distributed two-stage assembly flow-shop scheduling) into account, considering the real-time sharing of manufacturing resources (molds) driven by digitization. Combining order allocation decisions and shared resource scheduling problems with distributed production scheduling, this paper proposes an optimization model and selects an effective scheduling optimization method to provide theoretical and technical support for solving the manufacturing resource (molds) sharing and allocation problem in the distributed two-stage assembly manufacturing environment to maximize the overall profit and minimize the order delay penalty cost.

The remaining chapters of this paper are arranged as follows: Section 2 reviews the relevant literature. In Section 3, a distributed two-stage assembly scheduling model with mold sharing is proposed. Section 4 uses three intelligent optimization algorithms to solve and verify the model. Section 5 conducts the numerical simulation and analyzes the results. Section 6 presents the conclusions and suggestions for future work.

## 2. Literature Review

### 2.1. Distributed Two-Stage Assembly Scheduling

In recent years, the research on the distributed two-stage assembly scheduling problem (DTASP) has gradually increased. For example, Xiong and Xing [4] aimed at minimizing the maximum completion time and the weighted sum of the average completion time. A hybrid genetic algorithm combining a variable neighborhood search (VNS) algorithm and simplified variable neighborhood search (GA-RVNS) is proposed. Deng et al. [5] propose a mixed-integer linear programming model and a competitive meme algorithm (CMA) for the distributed two-stage assembly scheduling problem with the goal of minimizing the maximum completion time. The distributed two-stage assembly scheduling problem (DTASP) with an independent set time is an extension of the conventional distributed two-stage assembly scheduling problem. Zhang and Xiong [6] propose a "new memetic algorithm (MA)" based on social spider optimization (SSO), aiming at minimizing the total completion time. Yavari et al. [3] study an integrated model of the joint decision making of order acceptance and scheduling in the two-stage assembly scheduling problem, establish a mixed-integer linear programming model with the goal of profit maximization and propose an initiating method based on a semi-permutation genetic algorithm (SPGA). For the distributed two-stage hybrid flow shop scheduling problem (DTASP) which needs to consider the sequence-dependent preparation time, De-ming and Tian [7] propose an improved leapfrog algorithm aiming at minimizing the number of delayed jobs and the maximum completion time at the same time. Ya-ling and De-ming [8] propose a new "imperial competition cooperation algorithm (ICCA)" aiming to minimize the total delay time based on the imperial competition algorithm and historical evolutionary data. The algorithm is a multi-empire collaboration method based on the adaptive exchange between the weakest empire computing resources and the strongest empire search ability. These research studies focus on the optimal solutions of the DTASP in order allocation, production scheduling and other aspects, considering minimizing the maximum completion time, average completion time or total delay time. However, little attention has been placed on the DTASP with manufacturing resource sharing.

### 2.2. Mold Sharing

Some researchers have also studied mold sharing among multiple subjects. For example, Aghezzaf [9] discusses the capacity and warehouse management of the transfer

of a limited and very expensive number of molds between factories in a supply network and design a heuristic method based on linear programming (LP) combined with Lagrange relaxation and linear programming. Raa et al. [2] propose a mixed-integer linear programming model to solve the problem of exchanging molds between plastic product manufacturers to save costs and confirm that mold sharing can reduce about 10% of the total production cost through real-world cases. Zhang and Yang [10] believe that mold sharing can effectively reduce the cost of mold development and idle funds. From the perspective of social mass production, mold sharing improves the mold utilization rate of the whole industry, makes mold resources a public resource and thus maximizes the social income of mold resources. Research focused on the sharing of manufacturing resources (mold) is quite rare, especially research on the sharing of molds between multiple factories in distributed two-stage assembly scheduling settings. However, uncertain processing times are not covered in these research studies.

### 2.3. CPS in Distributed Scheduling

In virtual (digital) design and development, the key manufacturing resources are associated with the physical manufacturing resources in the distributed manufacturing environment in its whole lifecycle, which provides favorable conditions for the modeling and scheduling of manufacturing resource sharing [11]. Digital twin (DT) is a virtual representation of physical manufacturing resources, including various models of manufacturing resources and industrial interconnected big data. Cyber-physical systems (CPSs) are networked physical equipment intelligent systems that integrate computing, communication and control capabilities based on perceiving the physical environment. A CPS realizes the interaction between the computer process and physical equipment process through human–computer interaction interfaces. Operators can control physical entities distributed in different geographical locations in a remote, real-time, cooperative and intelligent way in networked space. A cyber-physical production system (CPPS) is primarily used to describe the CPS used in the production [12]. CPPSs, based on big data, network, and cloud computing, are widely accepted by the academic community because of their decision-making and intelligent resources. In order to improve the data management mechanism and make industrial big data better serve enterprise applications, Zhang et al. [11] introduce DT in the information layer and propose a general four-layer architecture of CPPs based on DT, namely a physical layer, network layer, virtual layer, and application layer.

In the academic field, the Internet of Things and CPS technology are mainly used in the research of intelligent manufacturing or intelligent factories, while the research literature on the combination of Internet of things, CPSs, and distributed production scheduling is relatively few. However, in recent years, some scholars have carried out relevant research. For example, Mahmoodjanloo et al. [13] use the real-time data obtained from the Internet of Things and CPPSs to deal with the dynamic scheduling problems of different production workshops composed of reconfigurable machine tools.

## 3. DTAFSP Model Design with Mold Sharing

### 3.1. Problem Description

The production process of large household appliances usually includes mold injection and final assembly, which belong to a typical two-stage assembly scheduling problem. Because large household appliance manufacturers usually configure multiple production plants and distribution warehouses nationwide or abroad, this problem is a distributed two-stage assembly scheduling problem. In each production plant, there are multiple mold injection lines and a final assembly line. Assuming that the Internet of Things and CPPs have been fully applied to each production base of a large household appliance group, the manufacturing resources (such as molds) of each base upload real-time data to the "cloud manufacturing resource pool" of the virtual space through the IoT and CPPs, as shown in Figure 1. All base factories rent molds from the "cloud manufacturing

resource pool", and the cost of mold leasing (transportation and use fees) is borne by the lessee. The sales orders generated by a group of sales companies are collected into the group's "order pool". The production cost and distance between the order destination and the distribution warehouse are considered during order allocation (order splitting). The distribution warehouse arranges the orders into production plans and allocates them to each factory for production. Mold capacity, logistics, and distribution cost, etc., need to be considered.

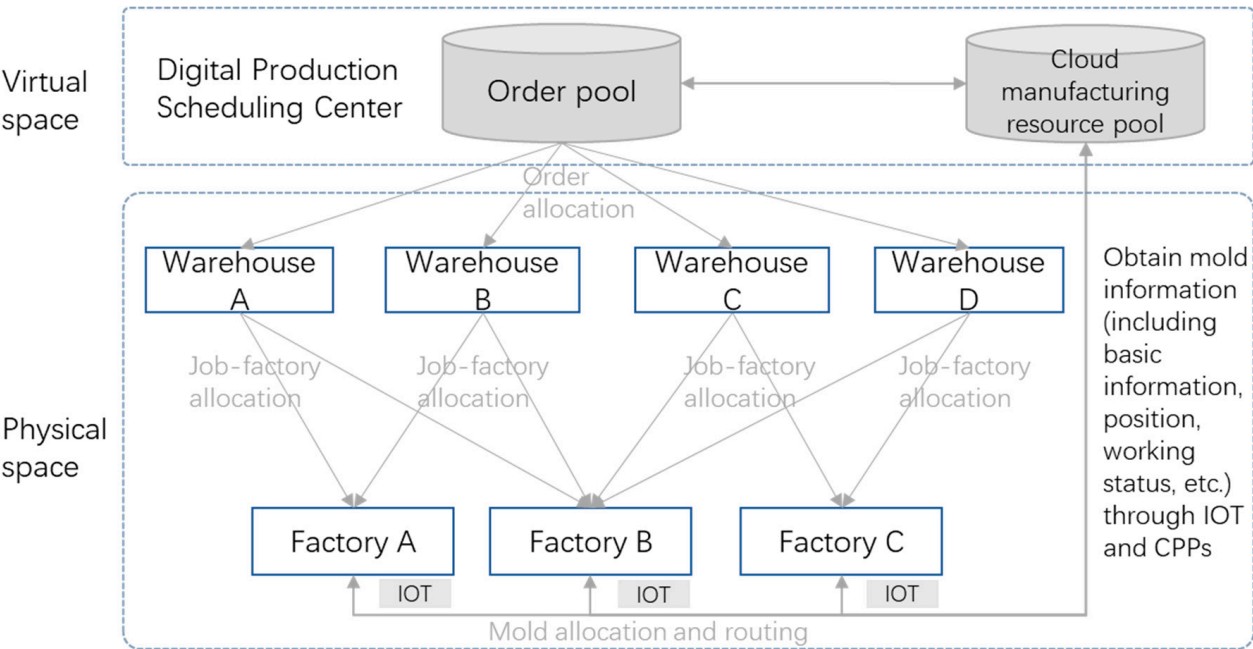

**Figure 1.** Distributed two-stage assembly scheduling with mold sharing.

The problem of sharing molds among multiple factories can be described as follows: Suppose n factories can produce certain types of household appliances. The set of factories is $P, p \in P$. The set of distribution warehouses is $W, w \in W$. The set of job is $J, j \in J$. The set of customers is $C, c \in C$. Molds can be transferred between factories, and there is a transfer time and a transfer cost.

Decision problem: in the distributed manufacturing environment with shared molds, the decision problem entails how to allocate orders and transfer molds among factories to maximize the total profit (i.e., order revenue minus production cost, logistics distribution cost, storage cost, warehouse incoming and outgoing processing cost, mold transfer cost, and delay penalty cost).

In order to facilitate data modeling and solutions, order virtual start and order virtual end nodes are introduced in each production plant. Virtual start and virtual end nodes are used as the starting point and end point of mold scheduling in each production plant. At the same time, it is assumed that all molds in the initial production stage of each factory are stored at the virtual start node of the order. After all orders in the factory are completed, all molds used need to be transferred to the virtual end node. In addition, the following assumptions are made:

(1) All factories and distribution warehouses have a common "order pool" and orders are homogeneous. Considering the delivery time and allocation cost of orders, different orders should be allocated to candidate distribution warehouses and factories.

(2) All factories are homogeneous and can produce any order in the order pool.

(3) The capacity of the same mold is the same and fixed. Each mold can only process one job at a time.

(4) The household appliance production line mainly includes two stages, namely injection molding and final assembly. Mold injection molding is in the first stage of the production line.

(5) When each factory starts, all molds are scheduled from the "cloud manufacturing resource pool", and all molds are scheduled from the beginning of job allocation. If there is no job assigned to factory *p* in the next stage, the molds in this factory *p* are returned to the mold warehouse or transferred to other factories which are assigned new jobs. The mold resource information in the "cloud manufacturing resource pool" is updated synchronously.

(6) In the process of order execution in each factory, except for sharing some molds, it is completely produced independently, and there is no timing constraint between orders in different factories.

(7) Different mold transfer strategies only affect the mold transfer cost and do not consider the possible impact on other costs of the production plant.

(8) Each factory has its own raw material procurement and sufficient inventory reserves Therefore, the transfer cost and time of raw materials and other materials are not considered.

(9) The transfer time of the mold from *i* to *j* is the same as that transferred from *j* to *i*.

(10) The setting time is negligible, the transportation time in factories is zero, the buffer size between the two stages is unlimited, and the mold and machine are continuously available.

*3.2. Symbol Definition*

The sets, parameters, and decision variables of the data model are defined as follows:

Set definition:

$P$: set of factories, $P = \left[ p_1, p_2, \ldots, p_f \right]$.

$J$: set of jobs $J = [J_1, J_2, \ldots, J_n]$.

$SKU$: set of SKUs, $SKU = [SKU_1, SKU_2, \ldots, SKU_m]$.

$T$: set of time unit, $T = [1, 2, \ldots, t]$.

Definition of decision variables:

$Y_{pj}^t$: the binary variable. If job *j* is allocated to factory *p* in time period *t*, it is 1; otherwise, it is 0.

$ct_{Aj}$: the completion time of job *j* on the assembly machine (phase II).

$T_j$: the delay time of job *j*.

$Q_{pj}^t$: the quantity of SKU *j* processed in factory *p* in time period *t*.

$U_{pjw}^t$: the quantity of SKUs transported from factory *p* to the distribution warehouse *w* in time period *t*.

$I_{wj}^t$: at the end of time period *t*, the inventory of SKU j in distribution warehouse *w*.

$V_{wjc}^t$: the quantity of SKU *j* transported from distribution warehouse *w* to customer *c* in time period *t*.

$X_{pjq}^t$: the binary variable. If the mold of job *j* moves from factory *p* to factory *q* in time period *t*, it is 1; otherwise, it is 0.

$L_{pj}^t$: the binary variable. If factory *p* is the last factory where the mold of job *j* is in time period *t*, it is 1; otherwise, it is 0.

$m_p^t$: the number of molds obtained by factory *p* in time period *t* (assuming that all molds are initially in factory 1).

$AT_{pjq}^t$: the arrival time of the mold of job *j* from factory *p* to factory *q* in time period *t*.

$ST_{pj}^t$: the starting time of job *j* in factory *p* in time period *t*.

Definition of model parameter:

$R_j$: the income from job *j*.

$w_j$: the delay penalty index for job *j*.

$D_j^t$: the due date of job *j* in time period *t*.

$wh$: the number of processing hours available in a time period.

$pt_{pj}$: the production time per SKU *j* in the first stage of plant *p*.

$at_{pj}$: the assembly time per SKU *j* in the second stage of plant *p*.

$pc_{pj}$: the total production cost of an SKU *j* in factory *p*.

$mt_{pjq}$: the transfer time required for injection mold of SKU *j* to be transferred from factory *p* to factory *q*.

$mc_{pjq}$: the transfer cost required for injection mold of SKU *j* to be transferred from factory *p* to factory *q*.

$tc_{pw}$: the transportation expenses required for transportation from factory *p* to distribution warehouse *w*.

$sc_w$: storage cost *w* required in the distribution warehouse.

$sv_w$: the storage capacity of distribution warehouse *w*.

$hc_w$: the processing cost of incoming/outgoing goods in distribution warehouse *w*.

$hv_w$: the loading and unloading capacity of incoming/outgoing goods in distribution warehouse *w*.

$tc_{wc}$: the cost of transportation from the distribution warehouse *w* to customer *c*.

$d_{cj}^t$: the quantity of SKU j required by customer *c* in time period *t*.

### 3.3. Mathematical Model

Objective function formula (1) takes profit maximization as the goal, and the profit is equal to the order revenue minus the production cost, transportation cost from factory to distribution warehouse and warehouse purchase processing cost, storage cost, transportation cost from distribution warehouse to customer and warehouse shipment processing cost, mold transfer cost, and order delay penalty cost.

$$maxmize\ z_1 = \sum_{t \in T} \sum_{j \in J} \sum_{p \in P} \sum_{c \in C} R_j d_{cj}^t Y_{pj}^t$$

$$- \sum_{t \in T} \sum_{j \in J} \left[ \sum_{p \in P} pc_{pj} Q_{pj}^t + \sum_{p \in P} \sum_{w \in W} (tc_{pw} + hc_w) U_{pjw}^t + \sum_{w \in W} sc_w I_{wj}^t + \sum_{w \in W} \sum_{c \in C} (hc_w + tc_{wc}) V_{wjc}^t \right.$$

$$\left. + \sum_{p \in P} \sum_{q \in P} mc_{pjq} X_{pjq}^t \right] - \sum_{j \in J} w_j T_j \tag{1}$$

s.t.

$$\sum_{p \in P} \left( \left( pt_{pj} + at_{pj} \right) Q_{pj}^t Y_{pj}^t + \sum_{q \in P} mt_{pjq} X_{pjq}^t \right) \le m_p^t wh \quad \forall j \in J, \forall t \in T \tag{2}$$

Including $m_p^t = \sum_{p \in P} Y_{pj}^t, \forall t \in T$

$$Q_{pj}^t = \sum_{w \in W} U_{pjw}^t \quad \forall t \in T, \forall j \in J, \forall p \in P \tag{3}$$

$$\sum_{p \in P} U_{pjw}^t - \sum_{c \in C} V_{wjc}^t + I_{wj}^{t-1} = I_{wj}^t \quad \forall t \in T, \forall j \in J, \forall w \in W \tag{4}$$

$$I_{wj}^0 = 0 \quad \forall j \in J, \forall w \in W \tag{5}$$

$$\sum_{j \in J} \sum_{p \in P} U_{pjw}^t + \sum_{j \in J} \sum_{c \in C} V_{wjc}^t \le hv_w \quad \forall t \in T, \forall w \in W \tag{6}$$

$$\sum_{j \in J} I_{wj}^t \le sv_w \quad \forall t \in T, \forall w \in W \tag{7}$$

$$V_{wjc}^t = d_{cj}^t \quad \forall t \in T, \forall j \in J, \forall c \in C \tag{8}$$

$$Q_{pj}^t, U_{pjw}^t, I_{wj}^t, V_{wjc}^t \ge 0 \quad \forall t \in T, \forall j \in J, \forall p \in P, \forall w \in W, \forall c \in C \tag{9}$$

$$(pt_{pj} + at_{pj}) Q_{pj}^t \le wh \left( \sum_{q \in P} X_{pjq}^t + L_{pj}^t \right) \quad \forall t \in T, \forall j \in J, \forall p \in P \tag{10}$$

$$\sum_{q \in P} X_{pjq}^t + L_{pj}^t \le 1 \quad \forall t \in T, \quad \forall j \in J, \forall p \in P \tag{11}$$

$$il_{pj} + \sum_{r \in P} X_{pjr}^1 = \sum_{q \in P} X_{pjq}^1 + L_{pj}^1 \quad \forall j \in J, \forall p \in P \tag{12}$$

$$L_{pj}^{t-1} + \sum_{r \in P} X_{pjr}^t = \sum_{q \in P} X_{pjq}^t + L_{pj}^t \quad \forall j \in J, \forall p \in P, \forall t \in T \tag{13}$$

$$X_{pjq}^t, L_{pj}^t \in \{0,1\} \quad \forall t \in T, \forall j \in J, \forall p, q \in P$$

$$\sum_{t \in T} \sum_{p \in P} \left( (pt_{pj} + at_{pj}) Q_{pj}^t Y_{pj}^t + \sum_{q \in P} mt_{pjq} X_{pjq}^t \right) = ct_{Aj} \quad \forall j \in J \tag{14}$$

$$\text{If } ct_{Aj} - \sum_{t \in T} D_j^t > 0 \quad \forall j \in J \text{ then } T_j = ct_{Aj} - \sum_{t \in T} D_j^t \quad \text{else } T_j = 0 \tag{15}$$

$$AT_{pjq}^t \leq ST_{pj}^t, \quad \forall j \in J, \forall p, q \in P, \forall t \in T \tag{16}$$

$$ct_{Aj} \geq 0, \qquad T_j \geq 0 \quad \forall j \in T$$

Equation (2) represents the capacity constraint of the factory (the capacity of the mold depends on the working hours of the mold in each period, while the capacity of the factory depends on the number of molds in the factory). Equation (3) shows that all SKUs produced in factory *p* are transported to distribution warehouse *w*. Equation (4) represents the in/out balance constraint of the distribution warehouse; that is, the inventory of distribution warehouse *w* at the end of the previous period plus the quantity of SKUs transported from factory *p* to distribution warehouse *w* is equal to the quantity of SKUs transported from distribution warehouse *w* to the customer plus the inventory of distribution warehouse *w* at the end of the current period. Equation (5) indicates that the initial inventory in the distribution warehouse is 0. Equation (6) represents the equilibrium constraint of the in/out cargo handling capacity of distribution warehouse *w*. Equation (7) represents the storage capacity constraint of distribution warehouse *w*. Equation (8) means that the needs of all customers are met in time. Equation (9) indicates that all quantity variables are positive numbers. Equation (10) indicates that production can be started only after the molds required by the factory arrive. Equation (11) ensures that the mold is in a certain period, either moving from factory p to factory q or staying in factory *p* until the next period (i.e., factory *p* is the last factory of the mold in the current period). Equations (12) and (13) are conservation constraints on the flow of molds: molds in a factory come from another factory or stay there from the previous time period and then leave another factory or stay for the next time period. Equation (14) stipulates that in time period *t*, the completion time, $ct_{Aj}$, of job *j* on the assembly machine is equal to the sum of its mold transfer time, $mt_{pjq}$, injection molding time on the mold, and production time on the assembly machine. Constraint (15) shows that the delay time of job *j* is equal to its completion time on the assembly machine minus the corresponding delivery time. Constraint (16) shows that the time when the mold arrives at the factory should be earlier than the time when the operation starts.

## 4. Solution Algorithm

### 4.1. Solution Parsing Heuristics

As shown in Table 1, in order to obtain a feasible and high-quality solution and improve the computational efficiency of the optimization algorithm, firstly, a set of heuristic programs for analyzing the solution by decoding is designed. The solution of each subsequent algorithm is fitted into the heuristic program to obtain the value of key variables for filtering. The heuristic program consists of the following main parts, including order–job allocation, order–factory allocation, factory–mold allocation, mold routing, and production scheduling, as well as the calculation of key constraints and main variables.

**Table 1.** The SPH procedure.

| |
|---|
| 1. Procedure |
| 2.     Set Parameters (***N_t, N_t1, N_t2, N_t3, N_job, N_sku, N_k, N_p, N_m, N, M***) |
| 3.     Input Solution Matrix |
| 4.     %%Job-SKU allocation%% |
| 5.     For *i* = 1: ***N_job*** |
| 6.         If ***J(i)*** is null and *i* ≤ *M* |
| 7.             J(*i*) = SKU(*i*)     % Assign product SKU(*i*) to job J(*i*) % |
| 8.         Else if *i* > *M* |
| 9.             J(*i*) = randi ([1 *M*],1)     % Randomly assign product ***SKU*** to job J(*i*) % |
| 10.         End if |
| 11.    End for |
| 12.    %%Job-factory allocation and initial scheduling%% |
| 13.    For *t* = 1: ***N_t*** |
| 14.      For *i* = 1: ***N_job*** |
| 15.         h(*i*)= randi ([1 *P*],1) % Randomly assign factory *p* to job *J(i)* % |
| 16.      End for |
| 17.         Calculate total quantity ***Q(p,j,t),U(p,j,w,t), I(w,j,t), V(w,j,c,t)***in period *t*. |
| 18.         Sum production quantity ***Q(p,j,t)*** = Sum transportation quantity ***U(p,j,w,t)***. |
| 19.         Sum transportation quantity ***V(w,j,c,t)*** = Sum order demand ***d(c,j,t)*** of customers. |
| 20.         Required production quantity ***Q(p,j,t)*** ≤ Total production capacity in period *t*. |
| 21.    End for |
| 22.    %%mold allocation and mold routing %% |
| 23.     ***wz_mj*** = [1111] % Initial position of 4 sets of molds |
| 24.     For *t* = 1: ***N_t*** |
| 25.       If (*t* = =1) |
| 26.         For *i* = 1: ***N_t1*** |
| 27.           If ***h(i)*** ≠ ***p1*** |
| 28.             Transfer mold from factory p1 to factory of ***h(i)***. |
| 29.             Record the new location of the factory where the mold is located. |
| 30.             Calculate the arrival time of mold ***AT(p,i,q)***. |
| 31.           End if |
| 32.         End for |
| 33.       End if |
| 34.       For *i* = ***N_t1*** + 1: ***N_job*** |
| 35.         If ***h(i)_mj*** = Empty     %if the factory of ***h(i)*** has no mold |
| 36.           For *p* = 1:***N_p*** |
| 37.             If ***wz_mj(p, t)*** ≠ Empty    %if the factory ***P(p)*** has mold in period t |
| 38.               Transfer mold from factory ***P(p)*** to factory of ***h(i)***. |
| 39.               Record the new location of the factory where the mold is located. |
| 40.               Calculate the arrival time of mold ***AT(p,i,q)***. |
| 41.             End if |
| 42.           End for |
| 43.         End if |
| 44.       End for |
| 45.     End for |
| 47.    %%Production Scheduling %% |
| 48.    For *t* = 1: ***N_t*** |
| 49.      For *j* = 1: ***N_job*** |
| 50.       For *p* = 1: ***N_p*** |
| 51.        If ***AT(p,j,q)*** ≤ ***ST(p,j)***   %The mold should arrive at the factory before production start% |
| 52.          Completion time ***CT(p,j,t)*** in factory *p* = Starting time + Production time in phase one and assembly time in phase two. |
| 53.        Else if ***CT(p,j,t)-D(j,t)*** > 0 |
| 54.          The order delay time ***T(j)*** = the completion time ***CT(p,j,t)***—the delivery time ***D(j,t)*** |
| 55.        End if |
| 56.         The order delay time ***T(j)*** = 0. |
| 57.       End for |
| 58.      End for |
| 59.    End for |
| 60. End Procedure |

### 4.2. Algorithm Design

Because genetic algorithms (GAs), the imperial competition algorithm (ICA), and the simulated annealing algorithm (SA), widely used in the fields of distributed production scheduling, have the characteristics of universality, flexibility, and strong robustness, they are suitable for parallel processing and can quickly obtain better optimization results when solving complex combinatorial optimization problems [14,15].

#### 4.2.1. Genetic Algorithm

A genetic algorithm (GA) starts from a population that may represent a potential feasible solution for a set of problems. Chromosomes containing gene characteristics form a population, and chromosomes are a mapping from traits to genes through specific coding [16]. The initial population follows the law of survival of the fittest, selects several individuals according to the value of the fitness function, and generates new individuals in line with the problem's solution through crossover and mutation operations, and it then selects the number of individuals in the population. After iteration, the near-optimal solution or optimal solution of the problem is finally generated. The processing flow of a genetic algorithm based on probabilistic evolution reversal is shown in Figure 2.

1. Encoding and decoding schemes

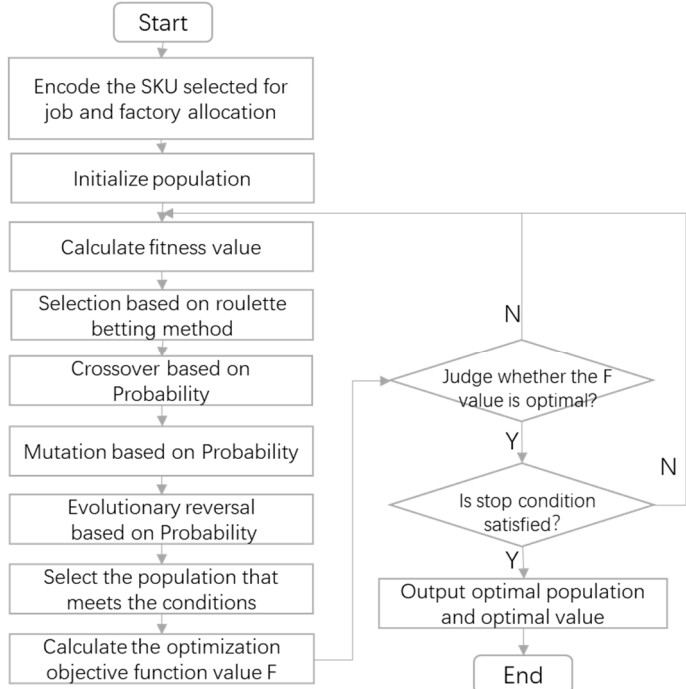

**Figure 2.** Flow chart of the GA.

When applying a genetic algorithm to solve the distributed production scheduling problem, a reasonable coding method is the key to solve production scheduling problems. Common coding methods include the following: binary coding, floating-point coding, symbol coding, real number coding, etc. A binary code is a symbol set composed of binary symbols 0 and 1. It has the following advantages: firstly, the operation of encoding and decoding is simple and easy; secondly, crossover, mutation, and other operations are easy to realize; thirdly, it conforms to the minimum character set coding principle; and fourthly, the algorithm is analyzed theoretically by using a pattern theorem. Since the mold transfer is determined by the factory to which the job is allocated for production, it is only necessary to encode the two subproblems of job–SKU allocation and job–factory allocation. Therefore, the coding string consists of two segments. The encoding string is $[h_1, h_2, \cdots, h_n; k_1, k_2, \cdots, k_{n-m}]$, where the first half of the string is the factory assigned

for a job, and $h_i \in \{1, 2, \ldots, f\}$ represents job $j$ assigned to factory $p_f$; the second half of the string is the SKU assigned to a job, and $k_j \in \{1, 2, \ldots, n - m\}$ represents the SKU $j$ assigned to job $j$. To decode the job sequence of DTASP based on mold sharing, interdependent decisions include job–SKU allocation, job–factory allocation, factory–mold allocation, mold routing, and production scheduling. The detailed decoding process can be referred to in Table 1.

2.  Fitness function

The fitness function reflects an individual's response to changes in the surrounding environment. The larger the fitness value, the better the characteristics of the chromosome. In a genetic algorithm (GA), the fitness function value is the only standard to evaluate the individual. The advantage of a genetic algorithm is that it can effectively improve the computing power of the algorithm and reduce the computing time [17]. The performance and convergence speed of a genetic algorithm depend on the rationality of the fitness function design.

Usually, the fitness function is designed around the objective function of the scheduling problem. In order to understand it more intuitively and conveniently, the fitness function can be directly equal to the objective function. The main goal of the DTASP based on mold sharing studied in this paper is to maximize the order profit under the condition of reduced delivery delays, so the profit function after deducting various costs is selected as the fitness function.

3.  Select operator

The core idea of the selection operation is "survival of the fittest". Good chromosomes have excellent genes, which can play a positive role in the iteration of subsequent genetic algorithms and guide the genetic algorithm to converge in the right direction. In this process, the genes of chromosomes with a high fitness value can be inherited by the next generation of population, cross-operate the optimized individuals, and then further optimize the new individuals; this allows good genomes to be preserved.

There are three commonly used selection methods, namely, the fitness-based proportional selection strategy, commonly known as the "roulette" selection method, the ranking selection method, and the best individual preservation strategy according to fitness.

The research of this paper adopts the fitness-based proportion selection strategy. Because this method is widely used, and no matter what kind of problem is presented, the implementation of this method is relatively simple and practical. The fitness value corresponding to the chromosome determines the probability of the individual being selected. We set the population size as $n$ and the fitness value of individual $i$ as $f_i$. The probability that the individual is selected in the cyclic iteration is as follows:

$$P_i = \frac{f_i}{\sum_{j=1}^{n} f_j} \tag{17}$$

It can be seen from formula (17) that the greater the proportion of individual fitness in the population, the greater the probability of being selected.

4.  Cross-operation

Crossover and mutation are the most important components of the genetic algorithm operation. Cross-operation refers to the process of exchanging some genes between two paired chromosomes in some way to produce two new individuals. Before cross-operation, individuals in the group are usually paired with each other according to the principle of random pairing. Common crossing methods include single-point crossing, two-point crossing, uniform crossing, arithmetic crossing, and so on. In uniform crossover, the genes at index $j$ of two chromosomes are exchanged with probability $P_c$. Relevant research shows that uniform crossover can better search the design space and maintain good information exchange. Therefore, this study selects the uniform crossover operation based on probability (as Figure 3).

5.    Mutation operation

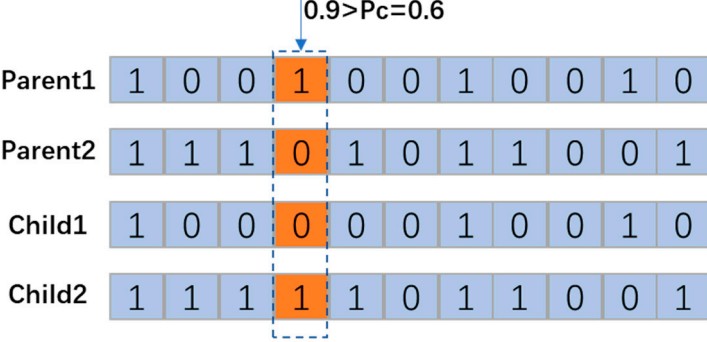

**Figure 3.** An illustration of gene crossover.

The stable genes of excellent parents are inherited by the chromosome genes of the new generation population, and the unstable genetic part also needs to be generated. The genetic difference between the old and new generations is realized through mutation operations. Mutation operations can ensure the continuous evolution of genes in the iterative process, increase the diversity of the population, and avoid the phenomenon of prematurity. Gene mutation in a genetic algorithm (GA) refers to replacing the gene values in the individual chromosome coding string with other allele values at this locus to form a new individual. Mutation operators suitable for binary coding include simple mutation, uniform mutation, boundary mutation, non-uniform mutation, and Gaussian approximation mutation. The mutation operator used in this study is that scanning each gene of the chromosome in turn and performing mutation operations on the value of a locus in the individual coding string with probability (as Figure 4).

6.    Evolutionary reversal

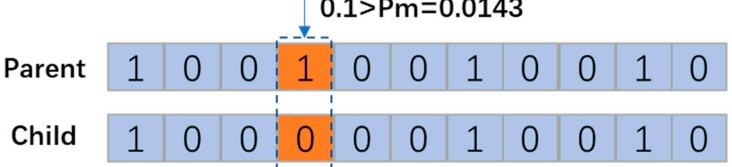

**Figure 4.** An illustration of gene mutation.

The GA has the disadvantage of an insufficient local search ability. After selection, crossover, and mutation operation, an "evolutionary reversal operator" based on probability is applied to enhance its local search capability. The reversal operator, also known as the inversion operator, is a genetic operator that allows for a more refined exploration of the solution space [17]. The reversal operator enables the genetic algorithm to jump out of the local minimum and quickly find the global optimal value [18]. "Evolution" here refers to the unidirectionality of the reversal operator. That is, after reversal, only the reversal with an improved fitness value can be accepted; otherwise, the reversal is invalid. Evolutionary reversal based on probability is the exchange of gene values at two randomly designated points on a chromosome with probability (as Figure 5).

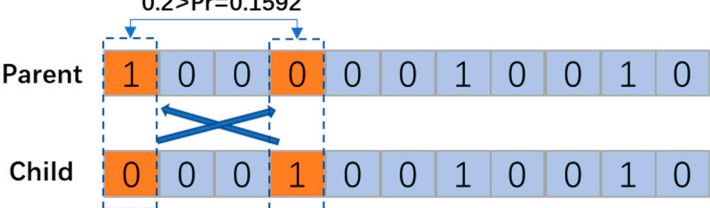

**Figure 5.** An illustration of evolutionary reversal.

The evolutionary reversal operator allows for a more refined exploration of the solution space, enabling an algorithm to make small, localized adjustments to a chromosome, while traditional GAs may struggle with fine-tuning solutions in the vicinity of local optima. Referring to Katoch et al. [17], the reversal operator helps the GA escape from local minima by flipping segments of the chromosome. If a segment of the chromosome is reversed, it may lead the algorithm out of a suboptimal area and towards potentially better solutions.

4.2.2. Simulated Annealing Algorithm

The simulated annealing algorithm (SA) is derived from the annealing principle of solid materials. The SA is a random optimization algorithm based on the Monte Carlo iterative solution strategy. Its original intention is based on the similarity between combinatorial optimization problems and solid material annealing processes. Starting from a higher initial temperature, the SA can find the global optimal solution of the objective function randomly in the solution space with the decreasing temperature parameters combined with the probability jump property. That is, it can jump out of the local optimal solution probabilistically and eventually approach the global optimal solution. The SA is a general optimization algorithm, which has a probabilistic global optimization performance in theory, so it has been widely used in distributed production scheduling domains, such as the distributed assembly replacement pipeline scheduling problem [19], distributed heterogeneous hybrid pipeline scheduling problem [20], distributed assembly displacement pipeline scheduling problem [21], and distributed arrangement flow-shop problem [22]. The processing flow of the simulated annealing algorithm is shown in Figure 6, and the specific steps are as follows:

1. Initialization: initial temperature *T0*, final temperature $T_{end}$, the state *x* of the initial solution (the starting point of the algorithm iteration), and the number of iterations of each t value $N_{iter}$.
2. For $n = 1, 2, \ldots, N_{iter}$, perform steps 3 to 6.
3. Generate new solutions $X'$: $X' = X + \Delta X$.
4. Calculate increment $\Delta f = f(X') - f(X)$, where $f(X)$ is the optimization objective.
5. If $\Delta f > 0$ (if looking for the minimum value, then $\Delta f < 0$), then accept $X'$ as the current solution; otherwise, accept $X'$ as the current solution with probability $exp\left(\frac{-\Delta f}{(kT0)}\right)$, where *k* is the Boltzmann constant and is usually set as *k* = 1 in practical problems.
6. If the stop conditions are met, then the current solution is output as the optimal solution, the current population is taken as the optimal population, and the program ends.
7. If *T0* decreases gradually, and $T0 > T_{end}$, then go to step 2.

Temperature management is one of the difficult problems to solve in the SA. In practical application, the feasibility of calculation time complexities and other issues need to be considered. The following cooling methods are usually adopted:

$$T = \alpha \times T0, \ \alpha \in (0, 1) \tag{18}$$

In Equation (18), in order to ensure a large search space, generally, the value of $\alpha$ is close to 1, such as 0.85 and 0.95.

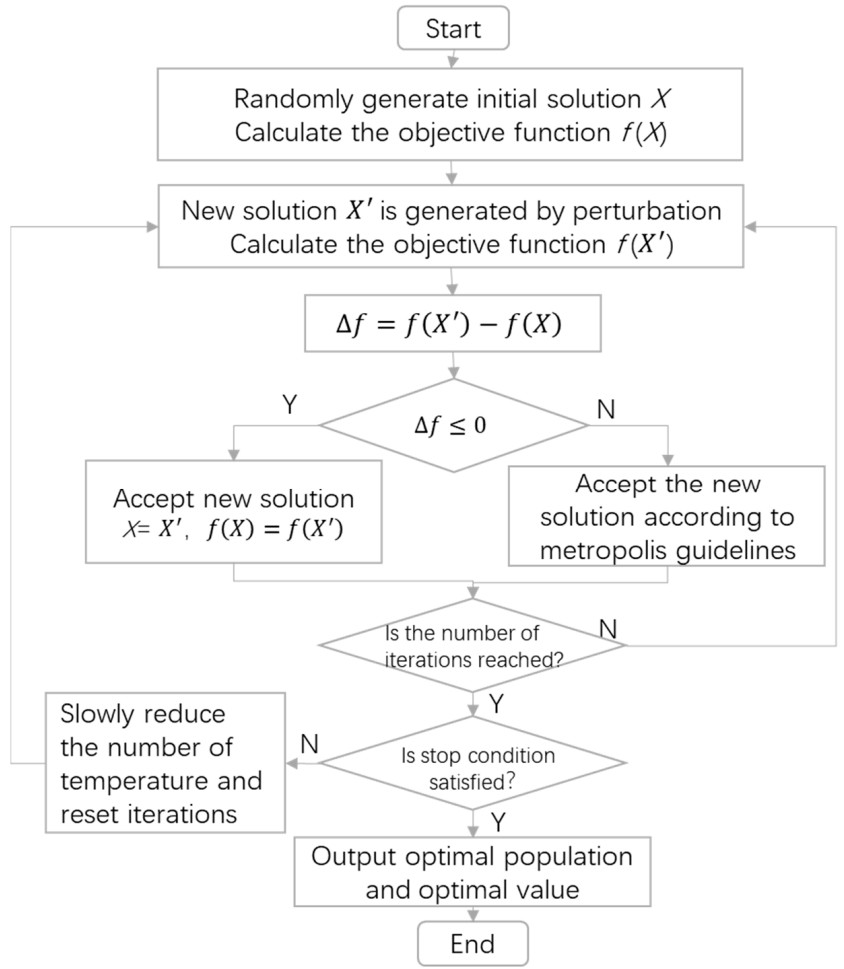

**Figure 6.** Flow chart of the SA.

### 4.2.3. Imperial Competition Algorithm

The imperial competition algorithm (ICA) is an intelligent optimization method initiated by imperial competition behavior. It belongs to a group-based random search optimization algorithm. Compared with the GA, the ICA has a high convergence accuracy, fast convergence speed, and strong global convergence. The solution space of the imperial competition algorithm is composed of individuals called countries. It divides individual countries into several subgroups, each of which is called an imperial group. Within each imperial group, through the imperial competition mechanism, one or more colonies in the weakest imperial group are transferred to other imperial groups, promoting the exchange of information between imperial groups. The ICA has been applied to distributed production scheduling problems, such as the distributed parallel machine scheduling problem in heterogeneous factories [23] and distributed two-stage assembly flow-shop scheduling problems [24]. The processing flow of the ICA is shown in Figure 7. The specific processing steps are as follows:

1.  Initialize country and generate Empire

The individual of the imperial competition algorithm (ICA) is the $country = [p_1, p_2, \ldots, p_n]$, which is equivalent to the chromosome in the genetic algorithm (GA). According to initial population size $N_{pop}$ and imperialist countries $N_{emp}$, we

generate the initial population and distribute colonies among imperialist countries. The power of each country is measured by the cost function as follows:

$$\text{cost} = f(\text{country}) = fitness([p_1, p_2, \ldots, p_n]) \tag{19}$$

Equation (19) equals the fitness function in this paper.

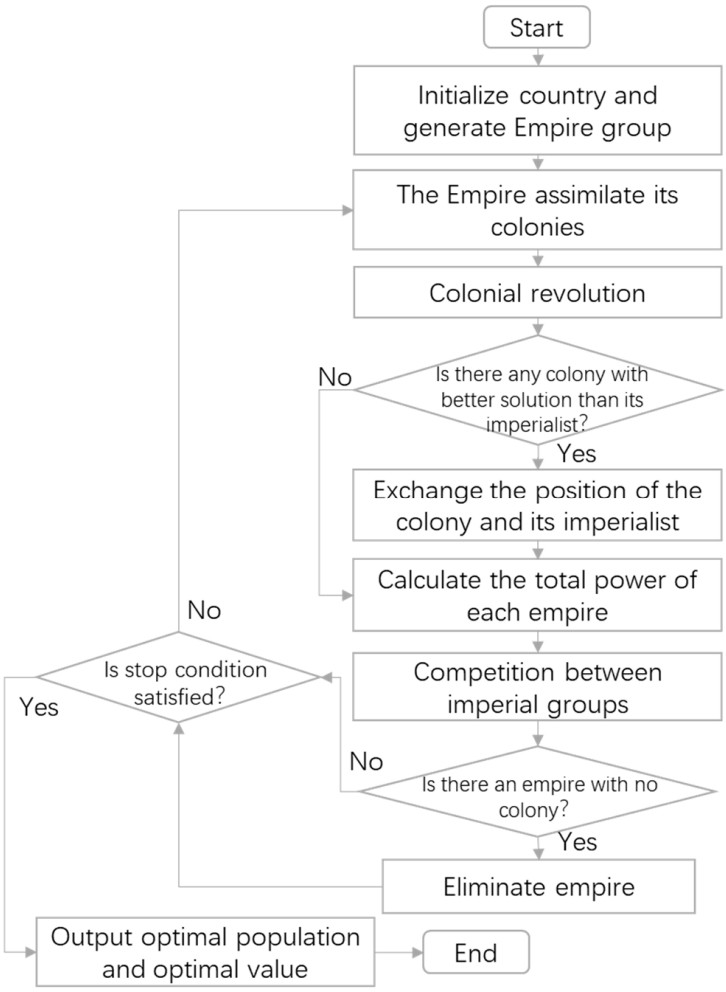

**Figure 7.** Flow chart of the ICA.

The division of colonies among imperialist countries is directly related to the power of each imperialist country. The cost of each imperialist country is defined as follows:

$$C_n = \max_{i=1,2,\ldots,N_{emp}} (C_i) - c_n \tag{20}$$

In Equation (20), $c_n$ is the cost of empire $n$. After calculating the cost of all imperialist countries, the standardized power of the empire is defined as follows:

$$P_n = \left| \frac{C_n}{\sum_{i=1}^{N_{emp}} C_i} \right| \tag{21}$$

The distribution of the original colonies depends on the power of the imperial group to which they belonged. The original colonial distribution formula is defined as follows:

$$N.C._n = round\{p_n \times N_{col}\} \tag{22}$$

In Equation (22), $N.C._n$ is the number of colonies owned by empire n. The larger the imperial group, the more colonies it has. At the same time, a weak imperial group has fewer colonies.

2. The empire assimilates its colonies

Assimilation refers to the process in which the empire extends its cultural customs and ideological models to the colonies in order to better control its colonies. The ICA simulates the assimilation process by moving colonies to their imperial group. This movement can be regarded as the movement of the solution in the solution space, which is essential during the transformation of the solution. Therefore, the new location of the colonies after assimilation is calculated as follows:

$$x_{Newcol} = x_{Oldcol} + \beta |x_{Oldcol} - x_{emp}| \tag{23}$$

In Equation (23), $\beta$ is the positive number that controls the assimilation operator.

3. Colonial revolution

Colonial revolution refers to a certain movement of the colony to make it closer to the optimal solution. But, revolution is not necessarily beneficial and may also lead to the internal consumption of resources, so it is impossible to achieve effective social change. At this stage, revolution percentage $P_{rev}$ of the colonies was randomly selected to carry out the revolution.

4. Exchange the positions of colonies and empires

After assimilation and revolution, the colonial countries move to a new position, so the cost function value of the colonial countries may be smaller than that of the empire; that is, the colonial power is greater. At this time, the positions of colonies and empires need to be exchanged. That is, the colonies become the empires of the imperial group, and the original empires become colonies.

5. Competition between imperial groups

The imperial competition mechanism simulates the process that the empire with stronger power gradually occupies and controls the colonies of the weaker power empire. Therefore, it is necessary to calculate the power of each empire, that is, the value of the total cost function. The total power of an imperial group consists of the dominant empire and the controlled colonies. The total cost of an imperial group is defined as follows:

$$T.C. = Cost(empire_n) + \xi mean\{Cost(colonies\ of\ empire_n)\} \tag{24}$$

In Equation (24), $T.C.$ is the total cost of imperial group $n$, and $\xi$ is a smaller positive number ($0 < \xi < 1$). The greater the value of $\xi$, the greater the influence of the colony on the imperial group.

6. Eliminate empire

The competition between imperial groups makes the powerful empires become more powerful by occupying the colonies in other imperial groups, while the number of colonies in the originally small imperial groups continues to decrease. When an empire completely loses its colonies, it dies. With the demise of the empire, there is only one empire left, and the algorithm terminates.

## 5. Numerical Simulation and Result Analysis

### 5.1. Parameters Setting

The simulation model uses the desensitization data of a large household appliance group enterprise for simulation. There are three remote factories, four sets of molds (shared and transferred among three factories), four remote distribution warehouses, and four remote customers. Table 2 shows the detailed parameter settings.

**Table 2.** Detailed parameter value range.

| Parameter | Parameter Range | Parameter | Parameter Range |
|---|---|---|---|
| $R_j$ | 122 | $sc_w$ | 5 |
| $w_j$ | 10 | $sv_w$ | 150 |
| $D_j^t$ | [3, 4, 1] | $hc_w$ | 2 |
| $wh$ | 40 | $hv_w$ | 150 |
| $pt_{pj}$ | [0.3 0.3 0.3 0.3 0.3; 0.4 0.4 0.4 0.4 0.4; 0.2 0.2 0.2 0.2 0.2] | $mt_{pjq}$ | [0 4 5; 4 0 3; 5 3 0] |
| $at_{pj}$ | [0.1 0.1 0.1 0.1 0.1; 0.1 0.1 0.1 0.1 0.1; 0.1 0.1 0.1 0.1 0.1] | $mc_{pjq}$ | [0 60 70; 60 0 65; 70 65 0] |
| $pc_{pj}$ | [96 98 97 96 99; 86 88 87 86 89; 85 85 86 85 87] | $tc_{pw}$ | [1 4 6 3; 4 1 7 3; 5 4 1 3] |
| $tc_{wc}$ | Because the relationship between the distribution warehouse and the customer is fixed (one-to-one), take the fixed value of 2 | $il_{pj}$ | [0 0 0 0 0; 0 0 0 0 0; 0 0 0 0 0] |
| $d_{cj}^t$ | t1: [5 0 0 0 0; 0 4 0 0 0; 0 0 5 0 0; 0 0 0 0 0] t2: [0 0 0 0 6; 4 0 0 0 0; 0 5 0 0 0; 0 0 0 5 0] t3: [0 0 0 0 0; 0 0 0 0 0; 0 0 0 0 0; 0 0 6 0 0] | | |

MATLAB is used for the algorithms' programming. The key parameters of the three algorithms are as shown Table 3. The number of iterations of the three algorithms is set to 30. The processor of a notebook computer running MATLAB is Lenovo MiiX 520 (made in China) with Intel (R) core (TM) i5-8250u CPU @ 1.60 GHz, 1.80 GHz, and 8 GB of RAM. In Table 3, the symbol $M_{Len}$ represents the Markov chain length.

**Table 3.** Key parameters of three algorithms.

| GA Parameters | | | | SA Parameters | | | ICA Parameters | | | | |
|---|---|---|---|---|---|---|---|---|---|---|---|
| $N_{pop}$ | $Pc$ | $Pm$ | $Pr$ | $M_{Len}$ | $T0$ | $\alpha$ | $N_{pop}$ | $N_{emp}$ | $\beta$ | $\xi$ | $P_{rev}$ |
| 100 | 0.9 | 0.1 | 0.2 | 100 | 10 | 0.85 | 100 | 10 | 1.5 | 0.2 | 0.2 |

### 5.2. Algorithms Comparison

The key of the distributed two-stage scheduling optimization with mold sharing is to maximize profit while minimizing the total order delivery delay time. Therefore, optimal profit and delivery delay time are two key indexes to measure the optimization effect of the model. Table 4 shows the solution results calculated by the three optimization algorithms under different mold transfer costs and transfer times. XmcXmt indicates the maximum mold transfer cost and time between different factories. For example, 5mc4mt means the highest mold transfer cost and the times between different factories are 5 and 4, respectively. Combinations in which mc takes the values {5, 25, 45, 75} and mt takes the values {4, 6, 8} are tested to verify whether there is a significant difference in the optimal profit and computation time obtained from the GA, ICA, and SA.

The Tukey multiple comparison method, also known Tukey's honestly significant difference (HSD) test, is a statistical technique used for comparing the means of three or more samples. The Tukey method involves calculating the differences between all possible pairs of sample means and comparing them to a specific critical value, which is suitable for analyzing differences between groups with equal sample sizes [25]. If the difference

between two sample means is greater than this critical value, it is considered that there is a significant difference between them.

**Table 4.** Comparison of optimization results of three algorithms under the same parameters.

| No. | Transfer Cost and Time | GA | | | ICA | | | SA | | |
|---|---|---|---|---|---|---|---|---|---|---|
| | | Optimal Profit | Delay Time (h) | Calculation Time (s) | Optimal Profit | Delay Time (h) | Calculation Time (s) | Optimal Profit | Delay Time (h) | Calculation Time (s) |
| 1 | 5mc4mt | 2235 | 0.3 h | 62.15 s | 2229 | 0.3 h | 61.31 s | 2269 | 0.3 h | 76.95 s |
| 2 | 25mc4mt | 2197 | 1.5 h | 67.59 s | 2191 | 1.5 h | 60.17 s | 2058 | 0.3 h | 75.2 s |
| 3 | 45mc4mt | 2085 | 1.5 h | 61.16 s | 2149 | 0.3 h | 60.33 s | 2032 | 1.5 h | 72.67 s |
| 4 | 65mc4mt | 2095 | 0.3 h | 62.98 s | 2095 | 0.3 h | 57.78 s | 2016 | 0 h | 73.9 s |
| 5 | 75mc4mt | 1860 | 0.3 h | 58.84 s | 2024 | 0.3 h | 59.73 s | 1932 | 0.3 h | 71.45 s |
| 6 | 25mc6mt | 2182 | 3 h | 62.43 s | 2142 | 3.5 h | 65.24 s | 2060 | 2.5 h | 75.804 s |
| 7 | 45mc6mt | 2082 | 3.5 h | 63.21 s | 2122 | 3 h | 60.53 s | 2000 | 3 h | 74.49 s |
| 8 | 25mc8mt | 2105 | 7.8 h | 62.04 s | 2122 | 9 h | 59.21 s | 1980 | 8.6 h | 76.3 s |
| 9 | 45mc8mt | 1988 | 5.3 h | 63.64 s | 2062 | 9 h | 61.09 s | 1937 | 8.2 h | 77.45 s |

Three independent samples are obtained by running the three algorithms 10 times under the same mold transfer cost and transfer time. The Tukey's HSD test was used to verify whether there are significant differences among the three independent samples. The calculation results are shown in Tables 5 and 6.

**Table 5.** Tukey multiple comparison results of the optimal profit.

| | Diff | Lwr | Upr | Adjusted *p*-Value |
|---|---|---|---|---|
| GA-ICA | 13.6 | −20.20229 | 47.40229 | 0.5846945 |
| GA-SA | −40.2 | −74.00229 | −6.39771 | 0.0173309 |
| ICA-SA | −53.8 | −87.60229 | −19.99771 | 0.0014349 |

**Table 6.** Tukey multiple comparison results of time consumed.

| | Diff | Lwr | Upr | Adjusted *p*-Value |
|---|---|---|---|---|
| GA-ICA | 5.763 | 3.226744 | 8.299256 | $1.63 \times e^{-05}$ |
| GA-SA | 21.960 | 19.423744 | −24.496256 | $0.00 \times e^{+00}$ |
| ICA-SA | 16.197 | 13.660744 | 18.733256 | $0.00 \times e^{+00}$ |

As Table 5 presents, referring to optimal profit, the mean difference between the ICA and GA is 13.6, with an adjusted *p*-value of 0.5846945. The difference between the ICA and GA is not significant at the 5% level of significance, while the mean difference between the SA and GA is 53.8, with an adjusted *p*-value of 0.0014349, which indicates that the difference between SA and GA is significant at the 5% level of significance. The difference between the SA and ICA is highly significant at the 5% level of significance since the mean difference between the SA and ICA is 87.6, with an adjusted *p*-value less than 0.001.

Referring to the time consumed, the results are illustrated in Table 6. Based on the adjusted *p*-values, there are significant differences between the means of the ICA-GA and SA-GA groups, while the difference between the SA and ICA is not statistically significant at the 95% family-wise confidence level.

It can be concluded that there are significant mean differences between the GA, ICA, and SA groups according to the Tukey multiple comparison test, especially with the SA group showing the most significant differences compared to the other two groups. A more visual result can be derived from Figures 8 and 9.

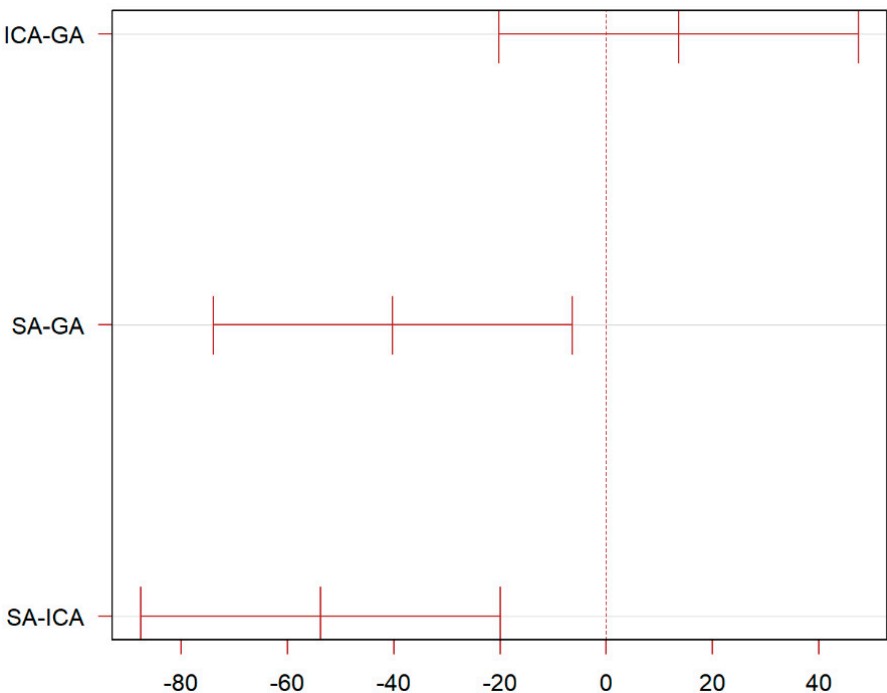

**Figure 8.** Tukey multiple comparisons of the optimal profit means.

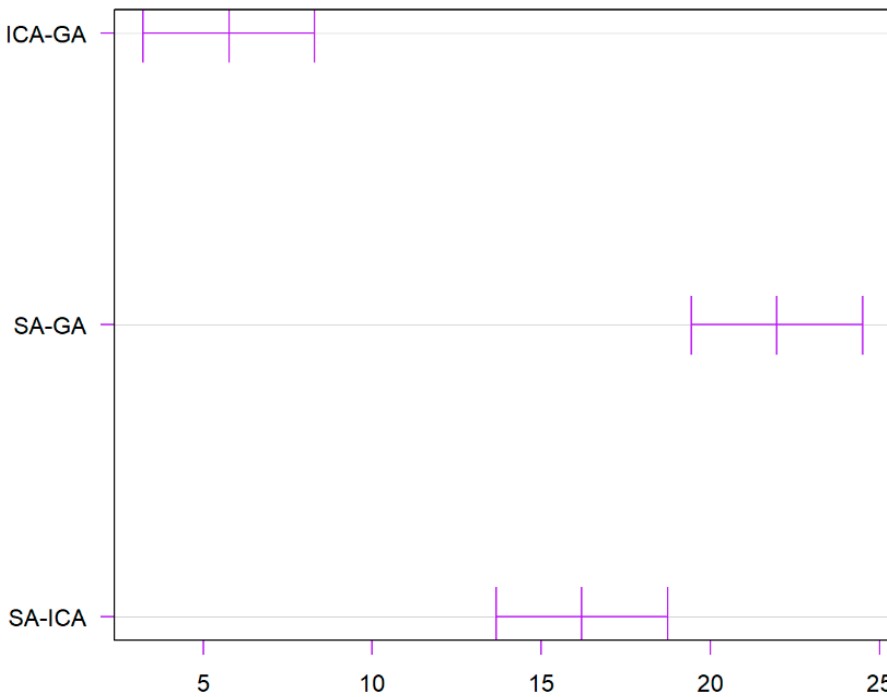

**Figure 9.** Tukey multiple comparisons of time consumed means.

From the box diagram shown Figure 10, the rank average value of the optimal profit of the genetic algorithm (GA) is relatively high, which meets the needs of solving the actual scheduling problem. In addition, the GA has the lowest rank average value of the calculated time consumed (as shown in Figure 11), so the computational performance is the best. Therefore, it is reasonable to choose the GA based on evolution reversal to solve the model.

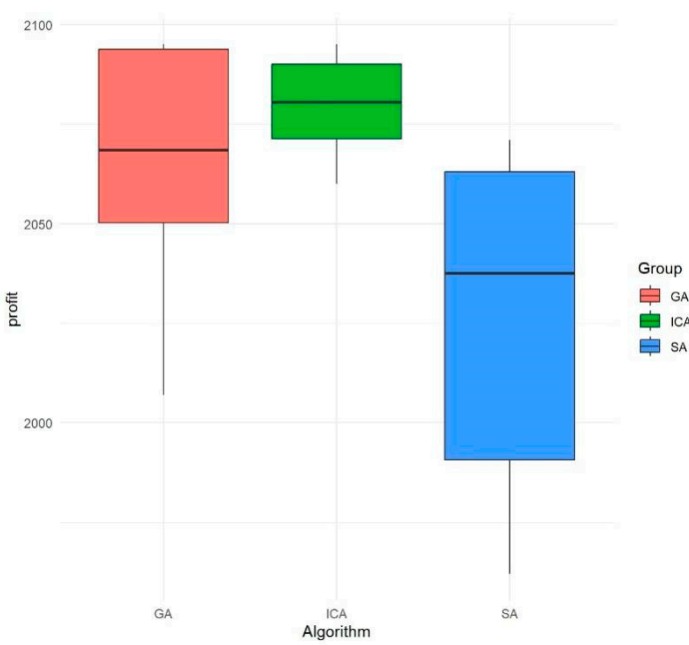

**Figure 10.** Boxplots of the Tukey comparison results (profit).

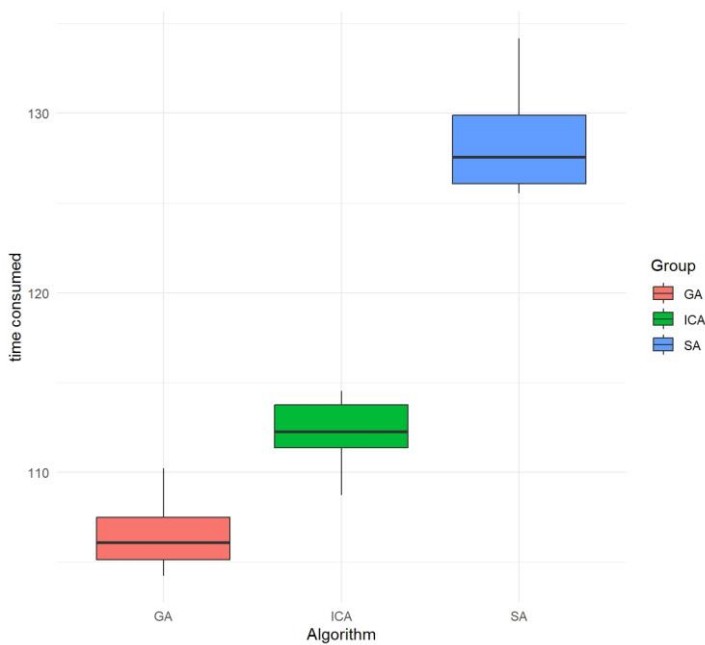

**Figure 11.** Boxplots of the Tukey comparison results (time).

*5.3. Results Analysis*

At present, large household appliance companies mainly allocate orders and assign factories based on the self-owned mold capacity of each factory, without considering the potential benefits that may be obtained by mold sharing and mold transfer across factories.

According to the comparison between the results before and after model optimization (Table 7), before optimization, that is, without mold sharing and transfer, the order delay time is 11.3 h, the total profit is 1152, and the gross profit margin on sales is 23.6% (note: the appliance group reported a 23% gross margin on sales in the first half of 2021, indicating that the simulated data are very close to the real data). However, under the same mold transfer cost and transfer time, the order delivery delay time is shortened to 0.3 h, according to the mold transfer parameter of 75mc4mt, the total profit is 1860, and the gross profit margin can reach 38.1% after optimization.

**Table 7.** Comparison of scheduling model before and after optimization.

| No. | Mold Transfer Cost and Time | Optimal Profit | Gross Margin | Delay Time (h) | Before and After Optimization |
|---|---|---|---|---|---|
| 1 | 5mc4mt | 2235 | 45.8% | 0.3 h | |
| 2 | 25mc4mt | 2197 | 45.0% | 1.5 h | |
| 3 | 45mc4mt | 2085 | 42.7% | 1.5 h | |
| 4 | 65mc4mt | 2095 | 42.9% | 0.3 h | |
| 5 | 75mc4mt | 1860 | 38.1% | 0.3h | After |
| 6 | 25mc6mt | 2182 | 42.2% | 3 h | |
| 7 | 45mc6mt | 2082 | 41.0% | 3.5 h | |
| 8 | 25mc8mt | 2105 | 40.6% | 7.8 h | |
| 9 | 45mc8mt | 1988 | 40.7% | 5.3 h | |
| 10 | 0mc0mt | 1152 | 23.6% | 11.3 h | Before |

Under the same mold transfer parameter of 75mc4mt, the optimal allocation results of job–factory allocation and mold distribution are as shown in Figure 12. In stage t1, there are three jobs, which are assigned to factories p2 and p3, and there are three molds transferred from the mold warehouse (assuming that the mold warehouse is in factory p1) to factories p2 and p3. In stage t2, there are four jobs, where two jobs are assigned to factory p2, and the other two jobs are assigned to factories p2 and p3. In stage t3, only one job is assigned to factory p3. If there is no job assigned to the factory in the next stage, the molds in this factory are returned to the mold warehouse or transferred to other factories which are assigned new jobs.

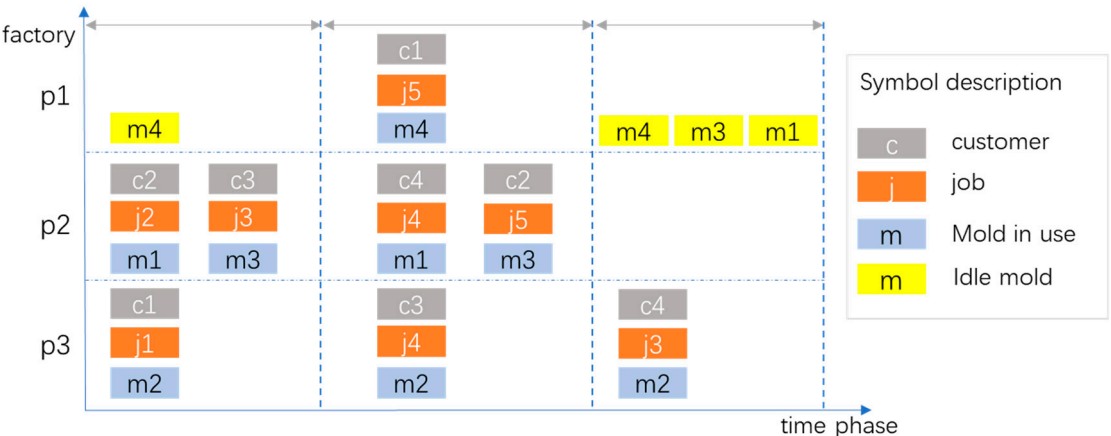

**Figure 12.** Optimal production scheduling when mold transfer parameter is 75mc4mt.

Obviously, in the case of mold sharing and transfer, the orders required by customers are assigned to the factory with the lowest comprehensive cost (including production cost, mold transfer cost, transportation cost, delay penalty cost, etc.). Compared with the previous mold where orders are assigned to the factory close to the customer location, the overall profit margin could be increased from 23.6% to 38.1% (by about 14.5%).

## 6. Conclusions and Future Recommendation

This paper discussed the sharing and scheduling of manufacturing resources (molds) in the distributed production settings of large household appliance manufacturing industries. The goal was to improve the total order profit and reduce the order delay time by sharing and allocating molds. Therefore, this paper first proposed a manufacturing resource (mold)-sharing mechanism based on the Internet of Things and a CPS, namely the "cloud manufacturing resource pool", to achieve the efficient allocation of manufacturing resources (mold). Then, factors such as order delivery delay, production cost, and mold transfer cost were considered, and a mixed integer programming model with the global goal of maximizing the total order profit was designed, which was solved and verified by a

genetic algorithm (GA) based on evolutionary reversal, the imperial competition algorithm (ICA) and the simulated annealing algorithm (SA). The experimental results showed that adopting the mold sharing and allocation mechanism based on the "cloud manufacturing resource pool" can effectively shorten the delay time of orders and increase the profit to distributed manufacturing enterprises.

However, this paper suffers some limitations. This paper assumes that there is no inventory of semi-finished products in the two-stage production. In addition, in order to improve the timeliness of the response to order delivery, the enterprise usually produces goods according to the forecast in the first stage of production and assembles them according to the order in the second stage. The uncertainties of forecast and demand are not included. Lastly, other heuristic algorithms, such as reinforce learning [26] and an improved genetic algorithm can be compared in a future study to explore a more effective way to settle this problem in a more complex situation.

**Author Contributions:** Conceptualization, C.M. and Y.L.; methodology, Y.H.; software, Y.L. and C.M.; validation, Y.H.; formal analysis, Y.L.; investigation, C.M.; resources, Y.H. and C.M.; data curation, Y.H.; writing—original draft preparation, Y.L. and C.M.; writing—review and editing, Y.L. and Y.H.; visualization, Y.L.; supervision, Y.H.; project administration, Y.L., C.M. and Y.H. All authors have read and agreed to the published version of the manuscript.

**Funding:** The research is funded by the Macau University of Science and Technology (Project No. FRG-22-107-MSB) and Huainan Normal University (Project No. 2022XJYB030).

**Data Availability Statement:** The data presented in this study are available upon request from the corresponding author. The data are not publicly available due to laboratory regulations.

**Acknowledgments:** Fundings from the Macau University of Science and Technology and Huainan Normal University are gratefully acknowledged.

**Conflicts of Interest:** The authors declare no conflict of interest.

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
