# Peer review of "An Internet of Things-Based Production Scheduling for Distributed Two-Stage Assembly Manufacturing with Mold Sharing"

_machines, doi:10.3390/machines12050310_

Round 1
Reviewer 1 Report
Comments and Suggestions for Authors
The maximum utilization of manufacturing resources (molds) in distributed two-stage assembly manufacturing is limited by the digitization and networking of manufacturing resources in different factory base. In the digital production scheduling center, order-factory allocation, factory-mold allocation, and mold routing can be performed centrally and efficiently. Therefore, in this paper, a manufacturing resource (molds) sharing mechanism based on Internet of things (IOT) and Cyber physical systems (CPS) is designed to realize the coordinated allocation of manufacturing resources (molds) and production scheduling. Considering the comprehensive utilization cost of manufacturing resources and the order delay penalty cost, a mixed-integer mathematical model is developed to optimize the cost structure and obtain a reasonable profit solution. Due to the existence of three NP-hard sub-problems, the proposed optimization model is very complex. Therefore, the genetic algorithm based on evolutionary reversal is used to solve the problem. The experimental results show that based on the digital coordinated production scheduling method, distributed two-stage assembly manufacturing with shared mold can effectively reduce the order delay time, and increase the total order profit, which can bring great potential benefits to distributed production enterprises.
The topic of the article is interesting, but the following issues deserve to be noticed.
1. The content of the article should be revised as the main innovation in the second page.
2. The GA appears to be in its most basic form. Is there any improvement in the algorithm?
3. There is no introduction to the need for SA and ICA.
4. Some studies in recent years considering the combination of reinforcement learning and GA should be mentioned. RL-GA: A reinforcement learning-based genetic algorithm for electromagnetic detection satellite scheduling problem. Swarm and Evolutionary Computation, 77:101236, 2023. reinforcement 808
Learning-assisted evolutionary algorithm: a survey and research opportunities. Swarm and Evolutionary Computation, 86:101517, 2024. 810.
5. The experimental section needs to add comparisons with state-of-the-art algorithms. Especially the reinforcement learning combined with EA.
6. The structure of the article needs to be adjusted. An overall framework of the study needs to be added. Problem description and modeling should also be adjusted to a visible position.
Comments on the Quality of English LanguageEnglish expression is fluent and clear.
Author Response
Comments and Suggestions for Authors
The maximum utilization of manufacturing resources (molds) in distributed two-stage assembly manufacturing is limited by the digitization and networking of manufacturing resources in different factory base. In the digital production scheduling center, order-factory allocation, factory-mold allocation, and mold routing can be performed centrally and efficiently. Therefore, in this paper, a manufacturing resource (molds) sharing mechanism based on Internet of things (IOT) and Cyber physical systems (CPS) is designed to realize the coordinated allocation of manufacturing resources (molds) and production scheduling. Considering the comprehensive utilization cost of manufacturing resources and the order delay penalty cost, a mixed-integer mathematical model is developed to optimize the cost structure and obtain a reasonable profit solution. Due to the existence of three NP-hard sub-problems, the proposed optimization model is very complex. Therefore, the genetic algorithm based on evolutionary reversal is used to solve the problem. The experimental results show that based on the digital coordinated production scheduling method, distributed two-stage assembly manufacturing with shared mold can effectively reduce the order delay time, and increase the total order profit, which can bring great potential benefits to distributed production enterprises.
The topic of the article is interesting, but the following issues deserve to be noticed.
Response: The authors really appreciate all your comments.
- The content of the article should be revised as the main innovation in the second page.
Response: Thanks very much for your advice. This part has been revised as the last paragraph of Page 1.
- The GA appears to be in its most basic form. Is there any improvement in the algorithm?
Response: Thanks very much for your comment. Although the basic form of GA has been used in this paper, the coding and decoding procedures have its own feature to fit this model and the optimization results are acceptable. In the future, the authors will further consider the improvement of this algorithm.
- There is no introduction to the need for SA and ICA.
Response: Thanks very much for your advice. The authors have added the need for SA and ICA as the first paragraph of Page 9 as follow.
“Because genetic algorithm (GA), imperial competition algorithm (ICA) and simulated annealing algorithm (SA), widely used in the fields of distributed production scheduling, have the characteristics of universality, flexibility, and strong robustness, they are suitable for parallel processing and can quickly obtain better optimization results when solving complex combinatorial optimization problems (Hosseini et al., 2014; Sangaiah & Khanduzi, 2022).”
- Some studies in recent years considering the combination of reinforcement learning and GA should be mentioned. RL-GA: A reinforcement learning-based genetic algorithm for electromagnetic detection satellite scheduling problem. Swarm and Evolutionary Computation, 77:101236, 2023. reinforcement 808
Learning-assisted evolutionary algorithm: a survey and research opportunities. Swarm and Evolutionary Computation, 86:101517, 2024. 810.
- The experimental section needs to add comparisons with state-of-the-art algorithms. Especially the reinforcement learning combined with EA.
Response: Thanks very much for your comment. In the future work, the comparison between reinforcement learning and other algorithms will be involved as stated as follows.
“Lastly, other heuristic algorithms, such as reinforce learning (Song et al. 2023) and improved genetic algorithm can be compared in the future study to explore a more effective way to settle this problem in a more complex situation.”
- The structure of the article needs to be adjusted. An overall framework of the study needs to be added. Problem description and modeling should also be adjusted to a visible position.
Response: Thanks very much for your comment. The authors have adjusted the structure of the article. The overall framework of the study has been added as the last paragraph of Section 1. Problem description and modeling have been revised as Section 3.
Reviewer 2 Report
Comments and Suggestions for Authors
1. The authors deliver a study with regards to IOT, mold sharing in a special type of scheduling problems, apply GA, SA and ICA to the problem with experimental results. The content is to the scope of the journal.
2. The organization of the draft is loosely coupled; for example, although section 3 mentions IOT, but nowhere can the reviewer find the solid linkage with the following sections. Another example is the formulation of the problem (as in section 4.2) does not relate to the following solving schemes since the problem is solvable with analytical solution and heuristics play no significant role in the example provided by the authors.
3. It is not clear why the authors add fuzzy variable Pij(theta) to address the uncertain issues in job processing time as well as mold transfer time (mode in p. 329 may be a typo) in the mixed integer program. Nowhere can the Pij(theta) be addressed in the subsequent sections. Or Pij(theta) is just a random variable?
4. For solving heuristics, the authors only express the common descriptions thereof, but without the crucial information about how the possible solutions are designed and how to solve the issues if the new solution is infeasible after mutation, crossover, perturbation, or position exchange operations.
5. The review is confused with the authors' use of CPS and CPP; are CPS and CPP the same or different? If different, why there is no mentioning about the meaning of CPP.
6. Fig 1 lacks explanation and a paragraph or two is suggested to supplement it.
7. Fig 2 is nicely drawn but without detail explanation like Fig 1.
8. For the shorthand representation of "transfer cost and time" column in Table 4, the reviewer can hardly comprehend the XXmcXmt. One possible reference of XXmcXmt is Table 2 (parameter value range), but some numbers used are nowhere found in Table 2.
9. According to Fig 10, the statistical test results are significantly different among the three heuristics. But it is preferred that the authors provide multi-comparison test to see which heuristic is best (or better).
10. The title of Section 6 is suggested to be changed to "Conclusions and Future Recommendation" to reflect the context thereof.
Author Response
Comments and Suggestions for Authors
- The authors deliver a study with regards to IOT, mold sharing in a special type of scheduling problems, apply GA, SA and ICA to the problem with experimental results. The content is to the scope of the journal.
Response: The authors thank you very much for providing the detailed comments.
- The organization of the draft is loosely coupled; for example, although section 3 mentions IOT, but nowhere can the reviewer find the solid linkage with the following sections. Another example is the formulation of the problem (as in section 4.2) does not relate to the following solving schemes since the problem is solvable with analytical solution and heuristics play no significant role in the example provided by the authors.
Response: Thanks very much for your comment. The authors have adjusted the structure of this paper. Section 3 describes the problem and IOT and CPPs act as the basic setting of proposed problem. The explanation is as the first paragraph of Section 3.1 as follows. The algorithms are described as Section 4. Section 4.1 The decoding algorithm for each of the algorithm in Section 4.2. As the first paragraph of Section 4.1, “each subsequent algorithm will be fitted into the heuristic program to obtain the value of key variables for filtering.”
“Assuming that the Internet of things and CPPs have been fully applied to each production base of a large household appliance group, the manufacturing resources (such as molds) of each base upload real-time data to the “cloud manufacturing resource pool” of the virtual space through IOT and CPPs, as shown in Figure 1. All base factories rent molds from the “cloud manufacturing resource pool”, and the costs of mold leasing (transportation and use fees) shall be borne by the lessee. The sales orders generated by the Group sales company will be collected into the group's “order pool”. The production cost and distance between the order destination and the distribution warehouse will be considered during order allocation (order splitting). The distribution warehouse will arrange the orders into production plans and allocate them to each factory for production.”
- It is not clear why the authors add fuzzy variable Pij(theta) to address the uncertain issues in job processing time as well as mold transfer time (mode in p. 329 may be a typo) in the mixed integer program. Nowhere can the Pij(theta) be addressed in the subsequent sections. Or Pij(theta) is just a random variable?
Response: Thanks very much for your comment. The authors have deleted the fuzzy variable.
- For solving heuristics, the authors only express the common descriptions thereof, but without the crucial information about how the possible solutions are designed and how to solve the issues if the new solution is infeasible after mutation, crossover, perturbation, or position exchange operations.
Response: Thanks very much for your comment. The decoding algorithm is presented as Section 4.1 Solution parsing heuristic. As the first paragraph of Section 4.1, “each subsequent algorithm will be fitted into the heuristic program to obtain the value of key variables for filtering.”
- The review is confused with the authors' use of CPS and CPP; are CPS and CPP the same or different? If different, why there is no mentioning about the meaning of CPP.
Response: Thanks very much for your comment. The difference is presented as the second paragraph of Page 3 as follows.
“Cyber-Physical Production System (CPPS) is primary used to describe CPS used in the production (Lee 2011). CPPS, based on big data, network, and cloud computing, is widely accepted by the academic community because of its decision-making and intelligent resources.”
- Fig 1 lacks explanation and a paragraph or two is suggested to supplement it.
- Fig 2 is nicely drawn but without detail explanation like Fig 1.
Response: Thanks very much for your comment. Considering the relativeness of the figures to the whole paper. The author has deleted Fig 1 &2.
- For the shorthand representation of "transfer cost and time" column in Table 4, the reviewer can hardly comprehend the XXmcXmt. One possible reference of XXmcXmt is Table 2 (parameter value range), but some numbers used are nowhere found in Table 2.
Response: Thanks very much for your comment. As the last paragraph of Page 16.
“XmcXmt indicates the mold transfer cost and time between different factories. For example, 5mc4mt means mold transfer cost and time are 5 and 4, respectively.”
- According to Fig 10, the statistical test results are significantly different among the three heuristics. But it is preferred that the authors provide multi-comparison test to see which heuristic is best (or better).
Response: Thanks very much for your comment. The comparison between the three algorithms are conducted as Fig. 9-10 of Page 17-18.
- The title of Section 6 is suggested to be changed to "Conclusions and Future Recommendation" to reflect the context thereof.
Response: Thanks very much for your comment. The title of Section 6 has been changed to “Conclusions and Future Recommendation”.
Round 2
Reviewer 1 Report
Comments and Suggestions for Authors
The authors have taken into account my previous comments. I accept this version.
Author Response
Thanks very much again for your valuable comments.
Reviewer 2 Report
Comments and Suggestions for Authors
1. The reviewer thanks the authors for very fast revision. However, the revision is too fast to be complete and thorough.
2. The reasons for resorting to heuristics rather than solving the mixed integer linear program are still not clarified.
3. The authors just delete the fuzzy variables without providing a solid reason. If there has no fuzziness/randomness in the model, following previous comment, the linear program is solvable with analytic solutions. Besides, the solutions should be provided as the comparison basis with heuristics presented by the authors.
4. For the solution infeasibility issues after mutation/crossover after GA, the authors still do not offer explanations to resolve them.
5. According to Table 2, the parameters of mc are in {0, 60, 65, 70}, but in Table 4, there have mc with {5 ,25, 45, 75}; why using the numbers not shown in Table 2? Similar situation can be found for mt.
6. Statistical multiple comparison is one kind of approach to compare many scenarios/alternatives (please refer to https://en.wikipedia.org/wiki/Multiple_comparisons_problem for details). In the study, since the authors intend to compare the three heuristics for their solution quality as well as computation time, it is suggested to report the statistical result professionally using multiple comparison statistical test method (i.e., Tukey or Duncan)
7. The notation in Fig 1 still uses wrong term--"model sharing" for "mold sharing."
Round 3
Reviewer 2 Report
Comments and Suggestions for Authors
No comment.
Author Response
The authors must express great thanks to the reviewer for your time and valuable advice to improve the manuscript.